# Prognostic and Therapeutic Role of CD15 and CD15s in Cancer

**DOI:** 10.3390/cancers14092203

**Published:** 2022-04-28

**Authors:** Wojciech Szlasa, Karol Wilk, Klaudia Knecht-Gurwin, Adam Gurwin, Anita Froń, Natalia Sauer, Wojciech Krajewski, Jolanta Saczko, Tomasz Szydełko, Julita Kulbacka, Bartosz Małkiewicz

**Affiliations:** 1Department of Minimally Invasive and Robotic Urology, University Center of Excellence in Urology, Wroclaw Medical University, 50-556 Wroclaw, Poland; karolwilk@me.com (K.W.); gurwin.adam@gmail.com (A.G.); anita.fron@student.umw.edu.pl (A.F.); wojciech.krajewski@umw.edu.pl (W.K.); tomasz.szydelko@umw.edu.pl (T.S.); 2Department of Dermatology, Venerology and Allergology, Faculty of Medicine, Wroclaw Medical University, 50-368 Wroclaw, Poland; klaudia.knecht@student.umw.edu.pl; 3Department of Drugs Form Technology, Faculty of Pharmacy, Wroclaw Medical University, 50-556 Wroclaw, Poland; natalia-sauer@outlook.com; 4Department of Molecular and Cellular Biology, Faculty of Pharmacy, Wroclaw Medical University, 50-556 Wroclaw, Poland; jolanta.saczko@umw.edu.pl (J.S.); julita.kulbacka@umw.edu.pl (J.K.)

**Keywords:** CD15, Lewis X, Lex, cancer, therapy

## Abstract

**Simple Summary:**

CD15 (Lewis X) is a typical myeloid antigen presented in myeloid and monocytic lineages of cells. This molecule interacts with E-, L- and P-selectins, which allows for adhesion with endothelial cells. CD15 is found on various cancer cells, including renal cancer, prostate and bladder cancers, acute leukaemias, hepatocellular carcinoma, breast cancer and melanoma cells. Its high expression can serve as a prognostic marker for patients and is a potentially valuable target for immunotherapy against cancer. Blockage of the antigen’s function results in reduced metastatic potential and it may be an immunotherapeutic target. CD15s is a sialyl derivative of CD15; however, unlike the high expression of CD15, which is a prognostic factor in Hodgkin lymphoma, CD15s relates to poor prognosis for patients. CD15 is considered a marker of cancer stem cells. This review presents a comprehensive description of the prognostic role of CD15 and CD15s and their use in anticancer therapy.

**Abstract:**

CD15 (Lewis X/Lex) is a fucosyl (3-fucosly-N-acetyl-lactosamine) moiety found on membrane proteins of various cancer cells. These cancers include renal cancer, prostate and bladder cancers, acute leukaemias, hepatocellular carcinoma, breast cancer and melanoma. The biological role of CD15 is interaction with E-, L- and P-selectins (adhesion molecules), allowing for adhesion with endothelial cells. In this way, cancer cells start to interact with the endothelia of blood vessels and consequently move out from the blood flow to the surrounding tissues. Blockage of the antigen’s function results in reduced metastatic potential. Moreover, the molecule may be a therapeutic target against cancer in monoclonal antibody-based therapies. CD15 may serve as a prognostic marker for patients and there are high hopes for its use in the immunotherapeutic treatment of tumours. CD15s is a sialyl derivative of CD15 that possesses its own unique characteristics. Its soluble form may act as a competitive inhibitor of the interaction of cancer cells with epithelial cells and thus disallow migration through the vessels. However, the prognostic relevance of CD15 and CD15s expression is very complex. This review presents a comprehensive description of the role of CD15 and CD15s in cancer development and metastasis and overviews its significance for clinical applications.

## 1. CD15 and CD15s: Expression, Structure and Homeostatic Function

CD15 (Lewis X) is a typical myeloid antigen found in the myeloid and monocytic lineages of cells. Depending on the cell origin, the expression of the antigen is determined by different isoforms of α1,3-fucosyltransferase. For instance, in mature granulocytes, it is the IX isoform, whereas in promyelocytes or monocytes it is isoform IV [1]. Moreover, the antigen is widely expressed among immature cells, including a range of CNS progenitors [2]. In general, tissues with high contents of hyaluronic acid seem to be positive for CD15 stain due to the high level of glycan content [3]. CD15 acts as a ligand for selectins in transendothelial migration. The molecule allows for the adhesion of a cell to blood vessel endothelial cells and the recruitment of circulating cells from the blood flow [4]. Further, cytokines such as TNFα, IL-6 and IL-1β activate CD62E (E-selectin) [5]. The molecules allow for the tighter adhesion of endothelial cells to the cell from blood flow. In the next step, the cell may migrate between the endothelial cells [6]. CD15 is abundant in normal cells and is considered the marker of myeloid cells. The molecule mediates neutrophil adhesion to dendritic cells [7]. CD15 also serves as a marker of granulocytes [8]. The antigen is also critical in the fetal development of the central nervous system and may be found in the interganglionic boundary (from the sulcus interstriatus to an area of the ventral margin of the caudate nucleus) [9].

CD15s is a sialyl form of CD15. The molecule may be found on a specific population of CD4^+^ cells. The CD15s expression pattern allows for discrimination between suppressive eTreg cells (CD15^+^) and cytokine-secreting non-Treg cells. CD4^+^CD15^+^ cell depletion from the CD4^+^ population evokes and enhances both anticancer (targeting NY-ESO-1) and antiviral (anti-CMV) immune responses [10]. The expression of both antigens (CD15 and CD15s) is similar due to a shared biosynthesis pathway (Figure 1). Interestingly the expression of α1,3-fucosyltransferase 7 determines the expression of the CD15s epitope [1]. Although the function of CD15 and CD15s remains connected to cell-to-cell adhesion and the extravasation process, the antigens are expressed on different cells and their expression correlates with different therapeutic outcomes for patients. Unlike CD15, CD15s may be expressed both on the surface of a cell and in serum in a soluble form. The latter may compete for interaction sites with E-selectin [11].

From the chemical perspective, CD15 is a fucosyl (3-fucosyl-N-acetyl-lactosamine) moiety, transferred and attached to membrane proteins by α1,3-fucosyltransferase 4. The same carbohydrate grouping may also be found on glycolipids and proteoglycans. Furthermore, the fucosyl moiety is thought to be responsible for migration, adhesion and immune response in tumour sites [12]. The carbohydrate structure of CD15 is found both on the plasma membrane and the cisterns of the Golgi apparatus [13]. Most of the studies of CD15 utilize the LeuM1 antibody to detect the antigen. Biosynthesis of CD15 involves the transfer of the fucosyl group to the Gal-GlcNAc saccharide. Interestingly, there is an alternative pathway of CD15 biosynthesis. It involves the initial biosynthesis of CD15s, which is a sialyl derivative of CD15. In the first step of the alternative pathway, sialyltransferase attaches an NeuNAc group to the Gal-Glc saccharide. Afterwards, the fucosyltransferase attaches a fucosyl group to GlcNAc in the core saccharide, thus forming CD15s [1]. The transition between CD15s and CD15 is catalyzed by sialidase, which releases the NeuNAc monosaccharide from CD15s, at the same time forming CD15 [7]. The increased expression of CD15 results from the increased activity of sialidase and not from de novo biosynthesis (Figure 1). 

## 2. Role of CD15 and CD15s in Neoplasms

Multiple studies discussing the role of CD15 and CD15s in cancer pathogenesis and prognosis have been published so far. Significant changes in cell surface carbohydrates were proved to accompany neoplastic transformation and they seem to be associated with tumour invasiveness and metastatic behaviour [14,15]. Overexpression of both CD15 and CD15s can be found on the surface of various types of cancer cells [16,17,18,19]. Various studies aim to correlate the expression of the molecule with the potential for tumour progression and survival of the patient. The molecule is especially important in tumour metastasis due to its role in the tight adhesion of cancer cells to blood vessel walls [20]. On the other hand, the expression of CD15s may be cellular and the antigen may also occur in soluble form. Soluble CD15s molecules in serum allow for competitive interaction with E-selectins and thus the inhibition of the escape of colorectal cancer cells from blood vessels [11]. Further paragraphs describe the prognostic role of CD15 and CD15s in neoplasms and the therapeutic approaches to targeting the antigen.

In general, the role of CD15 and CD15s in the neoplastic process relies on three main factors—allowing for adhesion to other cells and inducing changes in the conformation of cancer-associated membrane proteins. The first includes the effects of myeloid-derived suppressor cells (MDSCs) on cancer progression.

Elevated CD15 is associated with the adhesion of some cancer cells during the metastatic process, since CD15 located on cancer cells can bind by homophilic interaction with CD15 located on vascular endothelial cells. This process starts a heterophilic interaction with other cell adhesion molecules, including those of the selectin family, consisting of E-, L- and P-selectin. It was demonstrated that CD15 and CD15s play an important role in the adhesion of cancer cells to the endothelia of blood vessels [21,22,23,24]. (For reference, see Figure 2) P-selectin and E-selectin are located on the surface of endothelial cells. The initial step of cell adhesion (tethering), between CD15s and E-selectin, is crucial for cancer metastasis [25,26]. Furthermore, researchers have established that CD15s is the specific ligand of L-selectin, which, in turn, is constitutively expressed on leukocytes [27]. This results in the binding of leukocytes and cancer cells, which facilitates metastatic spread via the circulatory system. 

Another mechanism involving CD15 and CD15s in malignancy progression is their ability to change the structure of membrane-bound proteins (e.g., mucins), which may hide cancer cells from destructive NK cells [28]. The study by Hanski et al. proposed that increased expression of CD15s may be a result of the binding of the molecule with mucins into Sialyl-Lewis X-positive mucins [29]. Moreover, disintegration and remodelling mechanisms induce the reorganization of the cell membranes of tumours with high expression of CD15 antigens, allowing for the conferral of the adhesive properties of the cancer cells [30]. 

MDSCs were first described in 2007. They are a congregation of pathological myeloid precursors, activated due to chronic inflammation caused by growing tumours. MDSCs protect a tumour against the host immune system, providing suitable conditions for its growth [31,32]. CD15^+^ MDSCs are present in oral cancer tissues. Immunohistochemistry analysis of biopsy and resected specimens proved that decreasing their numbers with preoperative chemotherapy can improve prognoses [33]. Several factors regulate the functioning of MDSCs. One is the oxidized low-density lipoprotein (OxLDL), which regulates the function of endothelial cells and macrophage foam cells. The main OxLDL receptor is the lectin-like oxidized low-density lipoprotein receptor-1 (LOX-1) [34]. Human polymorphonuclear-myeloid derived suppressor cells (PMN-MDSCs), which are involved in NSCLC progression, can be determined by LOX-1 expression. Recently, it was found that LOX-1^+^ CD15^+^ PMN-MDSCs increase immune suppression and promote tumour expression. LOX-1^+^ CD15^+^ PMN-MDSCs can be useful in prognosis and recurrence after surgery. LOX-1^+^ CD15^+^ PMN-MDSC proportions are enhanced in cases of NSCLC and recurrence. All these findings were made through flow cytometric analysis of the peripheral blood of patients with NSCLC and health controls [35]. 

## 3. Prognostic Significance of CD15 and CD15s Expression in Various Neoplasms

Aberrant expression of CD15 was observed among various tumours. Not only were haematological malignancies stained positive for CD15 but also several solid tumours overexpressed CD15. For instance, adenocarcinomas express high levels of CD15 [13]. In 2016, Liang et al. performed a meta-analysis of 29 studies to establish the relationship between CD15s expression on the surface of malignant cells, cancer prognosis and clinicopathology [36]. The study proved that a high level of CD15s expression is significantly associated with lymphatic invasion, venous invasion, deep invasion, lymph node metastasis, distant metastasis, tumour stage, tumour recurrence and overall survival in cancer. The authors suggested CD15s as a new diagnostic and prognostic biomarker with the potential to become a therapeutic target in different types of cancer. Nevertheless, further studies are required to investigate the factors that caused significant heterogeneity in this meta-analysis. Each neoplasia may be characterized separately.

### 3.1. Gastrointestinal System Cancers

Nakagoe et al. analysed the expression of CD15s in a cohort of 101 patients with 0–II stage gastric cancer who underwent curative gastrectomy to clarify its prognostic value [37]. In 31 patients, high expression of the CD15s antigen was detected within tumours. These patients had shorter disease-free intervals (*p* < 0.0001) and worse disease-specific survival (HR 9.1 for high CD15s expression) than those with negative or low CD15s expression. The authors concluded that CD15s might serve as a prognostic factor in 0–II stage gastric cancer. Futamura et al. examined immunohistochemically the expression of CD15s in 245 patients with gastric cancer, which resulted in 135 (55%) positive cases [38]. Moreover, the occurrence of lymph node invasion, liver metastasis and stage III/IV tumours were significantly higher in CD15s-positive patients than in CD15s-negative ones (*p* < 0.01, *p* < 0.01, *p* = 0.028, respectively). The overall prognoses were also worse for patients with high CD15s expression (*p* = 0.019). An elevated expression of CD15 on gastric epithelial cells seems to be correlated with intestinal metaplasia, the precursor of gastric cancer [39]. 

Different changes were observed in oesophageal adenocarcinomas developed in Barret’s epithelium, in which the expression of CD15 was reported to be much lower than in non-Barret’s cancer [40]. However, the study cohort consisted of only 50 patients (17/50 Barret’s adenocarcinoma), which may have affected the reliability of the results. These observations seem to confirm the study of Faried et al., who demonstrated high expression of CD15 in 31% of patients (40/130) with oesophageal squamous cell carcinoma (non-Barret’s cancer) [41]. The authors found a strong correlation between occurrence of this antigen and worse TNM classification (*p* < 0.01), lymph node metastasis (*p* < 0.0001) and blood vessel invasion (*p* < 0.0001). The overall 5-year survival rate of these patients was significantly lower than the patients who were CD15-negative (10% vs. 66%, respectively, *p* < 0.0001).

The prognostic values of CD15 and CD15s in colorectal cancer have been fairly well documented over the years. The 5-year disease-free survival rates in a group of 132 patients were 58% and 89% for patients with CD15s-positive and CD15s-negative tumours, respectively (*p* < 0.001) [42]. Differences in the 5-year overall survival rates in this group were also significant—58% for CD15s-positive and 93% for CD15-negative patients (*p* < 0.001). Another study of 120 patients with colorectal cancer, of which 87 (72.5%) had high expression of CD15s, also resulted in a statistically significant difference between overall 5-year survival of CD15s-positive and CD15s-negative patients (61% vs. 81%, *p* < 0.05) [43]. Grabowski et al. divided 182 patients with colon cancer into 2 groups based on the expression intensity of CD15s on carcinoma cells, assessing the UICC stage [44]. Strong CD15s expression was detected in 103 patients, while it was weak in 79 patients. Strong expression of CD15s was associated with a reduction in the 5-year overall survival rate in UICC stage II (54% vs. 84%, *p* < 0.01) and stage III patients (35% vs. 86%, *p* < 0.01). Taking the value of CD15s expression in colorectal cancer into consideration, in 2012, Schiffmann et al. established a new scoring system as an easy tool to assess CD15s expression intensity [45]. Several studies aimed to determine the prevalence of CD15 on colorectal cancer cells and non-lesion, healthy colon cells. Shi et al., in one of the very first trials on this topic, demonstrated CD15 expression in 100% of colonic adenocarcinoma tissues examined [19]. However, the cohort included only 20 cases. In a more recent study, Portela et al. reported CD15s in 75% of colorectal cancer samples while in only 6.7% of healthy colon tissue samples obtained from the same patients [46]. Lastly, Jang et al. published the results of performing immunohistochemical staining for CD15 in 42 cases of colorectal carcinoma, 49 cases of tubular adenoma, 15 cases of hyperplastic polyp and 17 cases of non-neoplastic colon [47]. CD15 expression levels were significantly higher in colorectal carcinoma (48%) than in low grade tubular adenoma (23%), hyperplastic polyp (0%) and non-neoplastic colon (6%) (*p* < 0.05). Furthermore, CD15 expression was shown to progressively increase during cancer development and progression.

Hepatocellular carcinoma (HCC), the primary malignant disease of the liver, is different from other solid tumours because it is commonly associated with the occurrence of intrahepatic metastasis, which is a poor prognostic factor [48]. There is a correlation between CD15 expression and intrahepatic metastasis—69% vs. 30% (*p* < 0.02) occurrence in patients with CD15-positive HCC and CD15-negative HCC, respectively, although the difference in survival rate is not statistically significant [49]. CD15s expression in liver tissue was found to be not specific for HCC. The study of Fujiwara et al. revealed expression of this antigen in 53% of chronic hepatitis tissue specimens and 89% of pre-cirrhotic and cirrhotic tissue specimens [50]. Nevertheless, all of the HCC specimens had positive CD15s expression. Okada et al. observed either membranous or cytoplasmic expression of CD15s on HCC cells. The cytoplasmic-positive cells were well differentiated, while membrane-positive cells were less differentiated. Moreover, the authors demonstrated a positive correlation between tumour size and CD15s expression [51].

CD15 was shown to be a sensitive and specific marker of bile duct neoplasms; therefore, it may become a novel tool to differentiate dysplastic and neoplastic biliary cells from non-neoplastic tissue, which is a common diagnostic problem in indeterminate biliary stricture [52]. Kashiwagi et al. detected CD15s not only in gallbladder cancer cell cytoplasm (52%, 28/54) but also in cancer stroma (39%, 21/54) [53]. Stromal expression of CD15s was frequently associated with lymphatic invasion, venous invasion and lymph node metastasis (54%, 50% and 60%, respectively; *p* < 0.05), which are known factors of poor prognosis. 

### 3.2. Lung Cancer

In the study of Fukushima et al., out of the 92 lung cancer samples examined, CD15 and CD15s were detected in 42% (39 cases) and 57% (52 cases) [18]. Higher expression was reported in 54 lung adenocarcinomas—CD15 in 48% (26 cases) and CD15s in 72% (39 cases). These results indicate that CD15s antigens are useful markers for lung adenocarcinomas. The distinction between peripheral lung adenocarcinoma involving the pleura from pleural epithelial mesothelioma remains a serious problem in oncology. Comin et al. found CD15 to be the most specific marker in differentiating mesotheliomas from adenocarcinomas—only 4.5% of mesothelioma cases but 100% of adenocarcinoma cases expressed CD15 [54]. Mizuguchi et al. analyzed the clinical significance of serum CD15s concentrations as a predictor of lymph node metastasis, based on 272 patients with non-small cell lung cancer who underwent pulmonary resection [55]. The median CD15s serum concentrations were 44 U/mL for N2/N3 patients and 27 U/mL for N0/N1 patients. The 5-year survival rates of patients with concentrations of CD15s > 38 U/mL and those with lower concentrations were 32% and 69%, respectively (*p* < 0.0001), which suggests the potential usefulness of serum CD15s concentrations as staging markers and predictors. Likewise, CD15s expression on the surface of lung cancer cells was linked with a higher probability of post-operative distant metastasis and shorter overall survival [43]. 

Martín-Satué et al. proved the crucial role of CD15s in a lung adenocarcinoma metastasis process [21]. Approximately 85% of patients with lung cancer have non-small cell lung cancer (NSCLC) and 20–40% of these patients can develop brain metastasis [56,57]. It was proven, using three different methods and four human cancer lines, that fucosylated carbohydrate epitope CD15 and sialylated CD15s have a role in the developing brain tumour. Disruption of cerebral endothelial cell monolayers and cancer cell adhesion to cerebral endothelial cells are increased by overexpression of these epitopes. These findings demonstrate that these epitopes can be possibly used as metastasis biomarkers [58]. 

### 3.3. Breast Cancer

Another malignancy confirmed to express CD15 antigens on the surface of its cells is breast cancer [59,60]. Similar to lung adenocarcinoma, CD15 and CD15s have been established as key antigens in breast cancer progression and metastasis, enabling endothelial adhesion [17]. Sozzani et al. aimed to evaluate the prognostic value of CD15s in a long-term follow-up study and the results were different to those for other cancers [61]. A total of 127 consecutive patients with primary breast cancer were included in the trial. The median follow-up time was 140 months. CD15s antigen expression was found in 37 specimens (21%). Overall survival and disease-free survival were similar for CD15s-positive and CD15s-negative patients (62% vs. 60% and 59% vs. 56%, respectively). The expression of CD15s seems to be not associated with breast cancer survival. 

### 3.4. Haematological Malignancies

Classically, the antigen was used to distinguish between Hodgkin’ (positive) and non-Hodgkin lymphomas. Curiously, its expression pattern is not fully restricted to Hodgkin lymphoma, being expressed also in peripheral T-cell lymphomas and primary cutaneous anaplastic large cell lymphomas [13]. 

Hodgkin lymphoma was one of the first malignancies for which CD15 expression was demonstrated. The problems with immunophenotypic studies increase with the infiltration of tumours by CD15+ granulocytes. In these cases, it is sometimes hard to distinguish between Hodgkin and non-Hodgkin lymphoma [62]. In the 1980s, Hodgkin–Reed–Stemberg (HRS) cells were found to react with the LeuM1 antibody raised against the CD15 antigen [63]. The detection of CD15 on HRS cells has been used as a diagnostic marker of Hodgkin lymphoma for years [64]. Researchers found the expression of CD15 on HRS cells to be a favorable prognostic factor, while the expression of CD15s was correlated with poor prognosis [17,65]. Although the expression of CD15 is not entirely specific for HRS, it is rather sensitive—detected in approximately 80% of all classical Hodgkin lymphomas [66]. Proper diagnosis of Hodgkin lymphoma involves the identification of Reed–Sternberg cells (RSCs) in biopsy specimens. They are located against a rich background composed of cells such as lymphocytes, eosinophils and histiocytes which make them difficult to find [67]. CD15 can be used for fine-needle aspiration cytology (FNAC) identification of RSC. Cytopathological expression of CD15 occurs in 66.7% of cases. In addition, CD15 was found to be more effective on smears and cell blocks [68]. CD15 can also be detected via staining using an autoclave or a microwave. Positivity in a microwave test was 92% and in an autoclave was 50% [69]. Anti-CD15 antibodies were also introduced in the therapy of AML in a phase I clinical trial [70].

### 3.5. Gliomas

Gliomas are the most common malignant primary tumours of the brain and spinal cord and are mostly mortal. An experiment conducted in 2021 proved that differentiated glioma cells with high expression of CD15 in conditions of hypoxia undergo dedifferentiation into cancer stem cells. This study shows that CD15 can be a potential marker of malignant glioma progression [71]. 

### 3.6. Urological Malignancies

The expression of CD15 and CD15s was identified in urological malignancies. Sheinfeld et al. performed an immunohistochemical analysis which demonstrated that CD15 is not detected in the normal adult urothelium except for occasional umbrella cells [72]. Nevertheless, expression was identified in 100% of invasive urothelial cell carcinomas and 78% of carcinoma in situ cases, regardless of grade, stage, blood type or secretor status, which is in line with other studies [73]. The presence of CD15-positive cells in bladder specimens enhanced the detection of urothelial tumours, correctly identifying bladder cancers in 253/293 (86%) cases compared to 63% for cytology alone [72]. A recently published study by Ezeabikwa et al. resulted in an intriguing disclosure that CD15 is highly expressed on low-grade bladder cancer cell lines. In contrast, high-grade cell lines were associated with low or lack of expression (on normal bladder epithelial cells, CD15 was not expressed at all) [74]. Numahata et al. found CD15s expression in 70% urothelial bladder carcinomas and correlated it with lymph node invasion, blood-borne metastasis and a lower 5-year survival rate [23]. Researchers investigated the value of different urine markers in bladder cancer, which resulted in 86% median sensitivity (80–94%) and 73% median specificity (37–86%) for CD15 [75,76,77,78]. However, the sensitivity, with a median of 75% (68–79%), was worse in recurrent disease, while the median specificity increased to 82% (67–86%). CD15 was concluded in the systematic review to be one of the most promising urine markers of bladder cancer [79].

Overexpression of CD15 and CD15s antigens were also demonstrated in prostate cancer [80,81]. The upregulation of CD15s is associated with hormonal-resistant, aggressive disease [80]. High expression of CD15 and CD15s may influence prostate cancer progression through several mechanisms, including immune recognition by selectins or modifications of prostate cells mucins that enable cancer cells to evade destruction by NK cells [28]. In a recent study, CD15s expression on prostate cancer cells was found to be regulated by androgens [82]. This may be one of the factors explaining why androgens play a crucial role in the development and progression of prostate cancer and why androgen deprivation therapy is usually the first-line treatment in metastatic disease.

Despite all the findings above proving diagnostic and prognostic values in many cancers, the role of CD15 and CD15s antigens in RCC is still unclear and insufficiently studied. Several researchers confirmed the expression of these antigens on the surface of RCC cells [83,84,85,86]. However, their influence on RCC progression and prognosis is more controversial and complex than in other cancers. 

One of the first trials to investigate the presence of CD15 in RCC was enrolled by Cordon-Cardo et al. in the late 1980s [83]. It resulted in 76% (22/29) of CD15-positive specimens. Interestingly, the authors also analysed samples of 15 metastatic tumours, finding CD15 overexpression on occasional cells in only 20% of cases. Despite a small cohort, the percentage obtained is in line with more recent studies. Røge et al., based on results for 109 patients with different RCC types, reported CD15 expression in approximately 80% of cases, while Ordóñez et al. reported expression in 63% of 48 patients [84,85]. RCC cells were shown to express CD15, varying according to subtype. Pan et al. reported similar results based on the biggest cohort so far (328 cases) [86]. CD15 expression was found on 62% of clear cell RCCs (ccRCCs), 41% of papillary RCCs and 11% of chromophobe RCCs. The results were similar to findings of López et al. (130 cases), who demonstrated expression in 60% of ccRCCs, 56% of papillary RCCs and 0% of chromophobe RCCs [87]. The presented values seem to be lower than CD15 expression in RCC overall. Nevertheless, the difference may be made up by more recent results: Røge et al. found CD15-positive cells in 77% of ccRCCs and 85% of papillary RCCs (there were only 13 papillary RCC cases), while Wu et al. reported expression in 246/301 (82%) of ccRCC cases [84,88]. Several studies also found CD15 antigens on the proximal tubules of normal kidneys [83,88,89]. CD15 expression on RCC cells may have clinical value in itself. CD15 is considered one of the best markers to distinguish between RCC lung or pleura metastasis and mesothelioma—two malignancies presenting a wide variety of morphological patterns, confusing clinicians [85]. However, the conclusion is inconsistent with the previously cited Cordon-Cardo et al. findings of CD15 overexpression in metastatic tumours and needs further analysis [83]. Recently, CD15 prognostic value in ccRCC was demonstrated based on the biggest sample size so far [87]. The authors associated loss of CD15 expression with lymphatic invasion, vascular invasion, necrosis, higher tumour grade and a reduction in the 5-year overall survival rate from 37% to 26%. It was speculated that loss of CD15 expression indicates a poorer prognosis due to decreased ccRCC cell differentiation. In conclusion, CD15 expression is a good prognostic factor in ccRCC, although more data are needed to obtain convincing findings. 

The influence of CD15s on prognosis in RCC is debatable and so far the results presented have been contradictory at first glance. Nevertheless, the authors have agreed about CD15s overexpression in RCC. Koga et al. investigated the metastatic potential of RCC by intravenously injecting mice with four lines of human RCC cells [90]. One line had a significantly higher ability to produce pulmonary metastatic nodules, while the rest produced either few or no metastatic nodules. A flow cytometric analysis revealed that only this cell line demonstrated high CD15s expression. This finding indicates that CD15s possibly plays a critical role in the hematogenic metastasis of RCC. However, in the first trial with a human cohort, Fukushi et al. found CD15s expression to be a factor indicating good prognosis [43]. The authors of this study used FH6 monoclonal antibodies for CD15s detection. The interaction was positive in 47% of RCC samples. On the other hand, Tozawa et al., who used CSLEX1 monoclonal antibody for CD15s detection, demonstrated CD15s expression to be a factor indicating poor prognosis [91]. Expression, overall, was found in 100% of the examined cases, but higher expression intensity was correlated with higher tumour stage and grade, lymph node invasion, vascular invasion, metastasis and shorter tumour-free survival. In the most recent study, Kobayashi et al. used both FH6 and CSLEX1 monoclonal antibodies to explore the potential of expression of the CD15s antigen as a predictor of prognosis in 52 RCC cases [92]. The expression was positive for FH6 and CSLEX1 monoclonal antibodies in 54% (28/52) and 71% (37/52) of specimens, respectively. The expression status of CD15s in total did not impact disease progression or overall survival rate. However, CD15s antigens recognized using FH6 and those using CSLEX1 seemed to negatively affect disease progression and prognosis. Those detected by FH6 were suggested as a good prognosis factor, those detected by CSLEX1 a poor prognosis factor. The authors concluded that the combined use of FH6 and CSLEX1 monoclonal antibodies supports a powerful predictor for patients with unfavourable prognosis—those with CD15s expression associated with low FH6 and high CSLEX1. These monoclonal antibodies were found to react with different glycolipids. Moreover, it was shown that the sugar determinant of CD15s varies depending on tissue origin. Therefore, this might explain why the interaction between the CD15s sugar determinant of RCC and theoretically two CD15s-specific monoclonal antibodies were different and why the results of the aforementioned studies were initially opposite. However, more studies on the role of both CD15 and CD15s are required. 

## 4. Experimental Drugs Targeting CD15

Due to the important role of CD15 in cancer metastasis, the molecule is nowadays considered a potential target for cancer immunotherapy. Various drugs try to modulate both the expression of CD15 and E-selectin to disallow for cell adhesion to the endothelium and prevent the metastatic process. 

Several cancers have demonstrated the expression of CD15. These include papillary thyroid carcinoma, Hodgkin lymphoma, non-small cell lung cancer, oral cancer, glioma and breast cancer. Each neoplasm differed with respect to the expression of the antigen and thus to the prognosis for the patient and response to therapy. 

For the first time, in 2019, the expression of CD15, one of the cancer stem cell markers associated with patient prognosis, was proved in papillary thyroid carcinoma (PTC) via immunohistochemical staining of CSC markers in constructed tissue microarrays from PTC samples. The obtained results show that CD15 expression is associated with shorter progression-free survival (PFS) [93]. 

A great example was presented by Elola et al., who reported a promising attempt to prevent breast cancer cell metastasis by targeting the interaction between CD15 and adhesion molecules on endothelial cells [16]. The authors incubated MCF-7 breast cancer cells with HUVEC cells and analyzed the interaction of both cell types following incubation with mAbs against CD15. They revealed that mAbs might lyse the interaction between cells and thus prevent the spreading of cancer cells through the vessels. Based on the in vitro research, the authors proposed a model for preventing cancer metastasis by targeting CD15 [16]. Five percent of patients with breast cancer (BC) suffer from infiltration of the leptomeninges by metastatic carcinoma, also known as leptomeningeal carcinomatosis [94]. Circulating tumour cells (CTCs) appear in the blood as well as in cerebral spinal fluid (CSF) patients with breast cancer (BC) leptomeningeal metastasis (LM) [95]. In the 2017 experiment, CSF samples from patients with BCLM were analyzed using flow cytometry. For the first time, CD15 overexpression in CSF cancer floating cells was demonstrated, allowing the conclusion that CD15 is a potential prognostic biomarker of breast cancer metastasis risk, poor prognosis and tumour recurrence [96]. Another study was conducted on an MCF-7 cell line in subjects receiving anticancer therapy using IgM mAbs. Mordoh et al. proved that FC-2.15 mAbs against CD15 mediate complement cytotoxicity against tumours and reduce clonogenic capacity. The studies also showed the selectivity of the antibodies against cancer cells with no effect on normal bone marrow cells [59]. In human lung adenocarcinoma cells, CD15s antigens are involved in the adhesion process, and by targeting the antigen the metastatic potential of the cells might be decreased [21]. Tozawa et al. found an interesting side effect of cimetidine for patients suffering from RCC, namely, that the drug suppresses the expression of E-selectin on vascular endothelial cells, thus disallowing the adhesion and migration of cancer cells. The study suggests that cimetidine may inhibit the metastatic process in patients with RCC [87]. Jiang et al. proved that the adhesion of gastric cancer cells to the endothelium may be inhibited by andrographolide. The molecule blocks E-selectin expression and thus disallows adhesion via CD15–E-selectin contact. When cells treated with andrographolide are preincubated with E-selectin and CD15, the effect of the andrographolide is nullifying [97]. Moreover, S-nitrosocaptopril (CapNO) inhibited CD15s expression in a HT29 cell line. Lu et al. proved that the drug may be used for cancer metastatic chemoprevention [98]. 

Androgen deprivation therapy (ADT) acts on the CD15 antigen via the modulation of androgen receptors. The latter regulate the biosynthesis of associated glycans, such as sialyl-Tn, CD15, chondroitin sulfate and O-GlcNAc. Therefore, Munkley proved that ADT induces the modification of prostate cancer traits related to glycans on their cell membranes [82]. 

## 5. Clinical Trials Concerning CD15 Expression

Several clinical trials revealed the expression of the CD15 molecule in pathological conditions. Antibody-based therapies include the analysis of the CD15 molecule on various cancer cells. These are metastatic RCC, non-small cell lung cancer, MDS, ALL, AML, melanoma and colon cancer cells. Non-cancerous conditions include pulmonary embolism and deep vein thrombosis. All the clinical trials are summarized in Table 1.

In Table 1, there might be distinguished two main drugs in the use of which CD15 may play an important role—nivolumab and pembrolizumab. An example of the first is the phase 2 study NCT03595124, which aims to compare the treatment methods for a translocation renal cell carcinoma. The investigators are using nivolumab to treat metastatic renal cancers, Xp11.2 associated translocation RCC, stage III and IV RCCs and unresectable RCC. Even though CD15 is not directly involved in targeting the tumour, its expression is to be monitored on myeloid-derived stem cells, thus allowing for the monitoring of one of the most important components of cancer development and progression. The study is taking place in 256 locations and the investigators aim to record progression-free survival or death due to any cause, assessed up to 4 years. The study is estimated to end in 2031. Conversely, pembrolizumab was applied in the NCT04589013 study entitled “Prediction of Response to Treatment With Immunotherapy + Chemotherapy in Non-Small Cell Lung Cancer (PRINCE)”. In this case, the investigators aimed to evaluate a multiparametric test, including CD15, to predict the cumulative incidence of death or progression on treatment with pembrolizumab combined with chemotherapy in NSCLC. Estimated enrollment is about 350 participants and the study is predicted to be finished in November 2023. The efficacy of pembrolizumab will be assessed by computer tomography scans and the obtained data will be divided into groups characterized by the expression of six markers—PD-L1, CD8, FoxP3, PD1, CD163 and, especially interesting, CD15. The researchers will try to correlate the response to pembrolizumab with the expression of the mentioned antigens prior to the beginning of therapy. 

## 6. Conclusions

The results of studies conducted in recent years clearly indicate the important role of the moieties of membrane and adhesion molecules in the pathogenesis of various types of cancer. 

CD15 and CD15s expression is associated with lymphatic and venous invasion, lymph node metastasis, distant metastasis, tumour stage, tumour recurrence and overall survival in cancer. The antigens may serve as new diagnostic and prognostic biomarkers, with the potential to become therapeutic targets in different types of cancer. However, due to the complexity of the expression of the CD15–CD15s complex, further studies are required to adequately categorize the diagnostic parameters.

CD15 proved itself to be a potent target for cancer therapy. By acting on both cancer cells and immune myeloid-derived cells, clinicians may obtain a higher response to therapy and induce changes in tumour microenvironments that will inhibit cancer growth and progression. 

Current attempts should aim to improve the clinical application of anti-CD15 therapy and analyze the safety of the therapy. Otherwise, targeting CD15 and CD15s seems to be an interesting treatment option, as is proved by the progress made in successive experimental and clinical studies. 

## Figures and Tables

**Figure 1 cancers-14-02203-f001:**
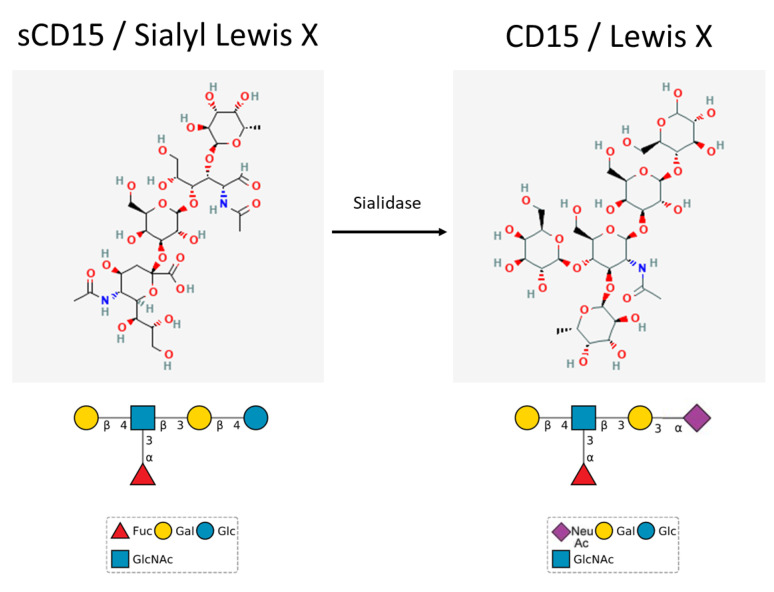
Structures of CD15 and CD15s, with the enzymatic reaction canalized by sialidase.

**Figure 2 cancers-14-02203-f002:**
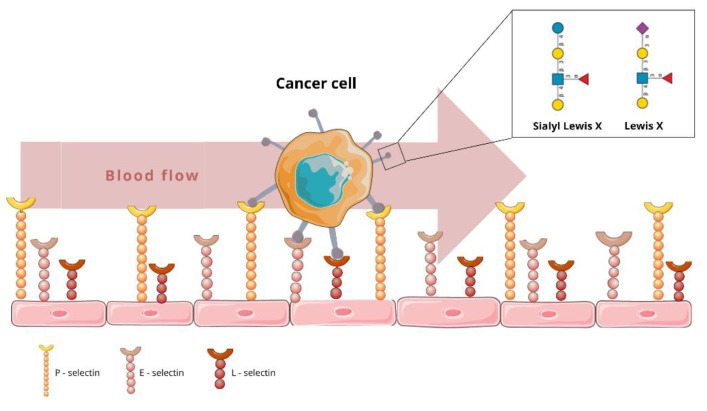
Cancer cell adhesion to endothelial cells by the Lewis X antigen interacting with E-, L- and P-selectins. The figure presents the first step in the transendothelial migration of cancer cells.

**Table 1 cancers-14-02203-t001:** Clinical trials involving the CD15 molecule.

Patient Number	Condition or Disease	Therapy Protocol	Analysed Antigens	Short Description	Recruitment Status	Ref.
40	Metastatic renal cell carcinoma, renal cell carcinoma associated with Xp11.2 translocations/TFE3 gene fusions, stage III renal cell cancer AJCC v8, stage IV renal cell cancer AJCC v8, unresectable renal cell carcinoma	CD drug: Axitinib Biological: Nivolumab	CD15, CD45, CD11b, CD33, CD14, HLA-DR, CE4, CD3, CD24, FoxP3, CD8, CD69, CD38, PD1, CD244, TIM3, CD4	Axitinib/nivolumab combination therapy vs. single-agent nivolumab for the treatment of TFE/translocation renal cell carcinoma (tRCC) across all age groups	Recruiting	NCT03595124
350	Non-small cell lung cancer	Drug: Pembrolizumab + chemotherapy	CD15, PD-L1, CD8, FoxP3, PD1, CD163,	Prediction of response to treatment with pembrolizumab + chemotherapy in non-small cell lung cancer	Recruiting	NCT04589013
18	Deep vein thrombosis, pulmonary embolism, cancer	Drug: Tinzaparin	CD15, CD24, CA19-9, TF, VEGF, TEPI	Thromboprophylaxis for patients undergoing surgical resection for colon cancer (PERI-OP)	Completed	NCT00967148
20	Myelodysplastic syndromes, MDS/MPN crossover syndromes	Drug: 5-azacytidine Drug: Decitabine	CD15, CD11b, CD14	5-azacitidine and decitabine epigenetic therapy for myeloid malignancies	Recruiting	NCT04187703
260	Acute myeloid leukemia, acute lymphoblastic leukemia, myelodysplastic syndrome	AlloHeme Test (ACROBAT)	CD15+, CD3+, CD33+, CD34+	Assessment of chimerism and relapse post bone marrow/hematopoietic cell transplant (HCT) using AlloHeme test (ACROBAT)	Not yet recruiting	NCT04635384
100	Colorectal carcinoma	Procedure: Fasting	CD15, CD3 CD4, CD8, CD19, CD45RA, CD62L, CD25, CD127, CD14, CD16, CD56, CD11b	Short-term fasting effects on chemotherapy toxicity and efficacy	Enrolling by invitation	NCT04247464
116	Acute myeloid leukemia	Drug: Transplants from 8/8-matched unrelated donors Drug: Transplants from family-mismatched/haploidentical donors	CD15, CD33, CD3	Transplantation from family-mismatched/haploidentical donors with matched unrelated donors in adult patients with acute myeloid leukemia	Unknown	NCT01751997
47	Advanced melanoma, recurrent melanoma, stage III cutaneous melanoma AJCC v7, stage IIIA cutaneous melanoma AJCC v7, stage IIIB cutaneous melanoma AJCC v7, stage IIIC cutaneous melanoma AJCC v7, stage IV cutaneous melanoma AJCC v6 and v7, unresectable melanoma	Biological: Pembrolizumab Biological: Talimogene laherparepvec	CD15, PD-L1, PD-1, CD80, CD86, FoxP3, CD68, PG-M1, DAKO, CD14	Talimogene laherparepvec (T-VEC) (NSC-785349) and MK-3475 (pembrolizumab) (NSC-776864) in patients with advanced melanoma who have progressed on anti-PD1/L1-based therapy	Active, not recruiting	NCT02965716
200	Renal cell carcinoma, clear-cell metastatic renal cell carcinoma	Biological: Nivolumab/ipilimumab	CD15, HLA-DR, CD11b, CD14, CD33, FoxP3, CD25, CD45RA, CD127, slan, CD1c, CD11c, CD123, CD141, CD303, ICOS, PD-1, PD-L1, CTLA-4, CD27, CD28, CD45RA, CD45RO, CD57, CD95, CD69, CD25, CD107a, TNF-α, IL-4, IL-17, IL-10,	Tailored immunotherapy approach with nivolumab in subjects with metastatic or advanced renal cell carcinoma (TITAN-RCC)	Active, not recruiting	NCT02917772

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
