# Peer review of "Prognostic and Therapeutic Role of CD15 and CD15s in Cancer"

_cancers, 2022, doi:10.3390/cancers14092203_

Round 1

Reviewer 1 Report

The review well describes the state of the art of CD15 and CD15s expression / overexpression in various human cancers. The article presents a comprehensive description of CD15 or CD15s (a sialyl derivative of CD15 ) as  prognostic markers in various different types of cancers and discusses the potential use of CD15 target-based immunotherapy for the treatment of cancer. In fact, CD15 and its syalilated form CD15s are ligands of the adhesion molecules E, P and L selectin and are involved in metastatic process for cancer cells.  The section “4. Experimental medicine involving CD15” well describes the latest news in this regard.  Moreover, the review focuses on the role of CD15 and CD15s in renal cell carcinoma (RCC). It is of interest considering that the role of CD15 and CD15s antigens in RCC is still poorly studied and the influence of those molecules on RCC progression and prognosis is more controversial and complex than in other cancers. The review describes the main studies on this field. It supports the concept that CD15s overexpression correlate with a higher metastatic potential of RCC and suggest CD15 as a potential target for renal cancer therapy. Interestingly, some apparent contradictory results in the literature on CD15s expression on RCC are explained by the authors by the different monoclonal antibody used to reveal this antigen. 

In general, the review is interesting but requires some adjustments to facilitate reading and comprehension. 

  1. Both "Simple Summary" and "Abstract" need to be remodulate. Authors should provide information on the physiological role played by CD15 and CD15s molecules before describing expression / overexpression on tumors and their involvement in metastatic processes. Moreover, since the review focus on the role of these molecules on CCR, this should be mentioned and discussed.  
  2. In the section “1. CD15 and Cd15s ”physiological” function” a brief description of the physiological role of CD15 and CD15s is required. Moreover, all the parts regarding the role of these molecules in cancers should be moved in the specific paragraph (see point 3). 
  3. in the section ”2.Role of CD15 and CD15s in neoplasms” should be reported all the information regarding the CD15 and CD15s expression/overexpression and its significance in hematological and solid tumors. In particular the phrases in paragraph 1 lines 70-82 and 88-89 should be rephrased and inserted in this section. 
  4. the bibliography will have to be rearranged according to the changes to the text that will be made. 

Typing errors

Line 85: further instead furhter 

Figure 1: CD15s instead sCD15

Line 237: one dot instead ..

Line 432: seems instead seemss" 

Author Response

cancers-1652626

Response letter to the Reviewer #1 Report

We thank the Reviewer for encouraging feedback and appreciate the insightful comments and suggestions.

Below, we provide a point-by-point response to each of the reviewer’s comments.

All changes in the manuscript were highlighted in yellow for clarity.

We hope that the introduced revisions significantly improve the quality of this review and qualify it for further editorial stages.

Sincerely,

Authors

Major aspects:

  1. Both "Simple Summary" and "Abstract" need to be remodulate. Authors should provide information on the physiological role played by CD15 and CD15s molecules before describing expression / overexpression on tumors and their involvement in metastatic processes. Moreover, since the review focus on the role of these molecules on CCR, this should be mentioned and discussed.

Response: Thank you for this valuable suggestion.  Simple summary and Abstract have been adapted to reflect major changes to the text of the main manuscript. We have changed both sections to show the complexity of the CD15 / CD15s antigens, but also some general information about them.

  1. In the section “1. CD15 and Cd15s ”physiological” function” a brief description of the physiological role of CD15 and CD15s is required. Moreover, all the parts regarding the role of these molecules in cancers should be moved in the specific paragraph (see point 3).

Response: Thank you for the comment. We moved the parts of section 1 concerning cancer to section 2. Moreover, we highlighted the physiological role of CD15 and now paragraph 2 starts with the required information about the function of CD15 and CD15s.

  1. In the section ”2.Role of CD15 and CD15s in neoplasms” should be reported all the information regarding the CD15 and CD15s expression/overexpression and its significance

Response: Thank you very much for this valuable comment.  We have moved all the information concerning the mechanism of cancer progression / expression to the general section 2. Additionally, we have created section 3 in which we describe the expression and its significance for various groups of tumors. Now, we hope the text gained in clarity and is properly structured.

  1. The bibliography will have to be rearranged according to the changes to the text that will be made.

Response: Thank you for your comment. We have changed the bibliography to match the new layout of the text.

Minor aspects:

Typing errors

  1. Line 85: further instead furhter
  2. Figure 1: CD15s instead sCD1
  3. Line 237: one dot instead ..
  4. Line 432: seems instead seemss"

Response: Thank you for your comments. All indicated errors have been properly corrected.

Reviewer 2 Report

In this review titled “CD15 and CD15s in the Anticancer Therapy”, Szlasa and colleagues have provided an overview of the expression and prognostic significance of CD15 in various cancers along with a list of clinical trials. The topic of the review is of high interest and importance to the field. However, the manuscript’s clarity and flow should be further improved before it can be considered for publication. Mainly, it lacks novel content and thought-provoking discussion in all the sections. While the title highlights the role of CD15 in anti-cancer therapy, the majority of the review describes its prognostic relevance of it in various cancers. Unfortunately, the review in its current form needs significant improvements in the content and flow.  

Author Response

cancers-1652626

Response letter to the Reviewer #2 Report

We thank the Reviewer for encouraging feedback and appreciate the insightful comments and suggestions.

Below, we provide a point-by-point response to each of the reviewer’s comments.

All changes in the manuscript were highlighted in yellow for clarity.

We hope that the introduced revisions significantly improve the quality of this review and qualify it for further editorial stages.

Sincerely,

Authors

Major aspects: 

  1. The manuscript’s clarity and flow should be further improved before it can be considered for publication.

Response: Thank you for this very valuable suggestion. As for the flow and clarity, we have added several linking fragments to the text and reorganized it thoroughly, to make it more comprehensive. Moreover, we have divided the text into subparagraphs to make it clearer for the reader and avoid the chaos from the initial version of the article. Now, each type of cancer has its own subsection, which structures the main text. We’ve corrected the clarity of the language used in the text. Additionally, the manuscript underwent extensive English correction by the native speaker.

  1. Mainly, it lacks novel content and thought-provoking discussion in all the sections.

Response: Thank you for the comment. In revised version, we’ve added more information about the most interesting and controversial parts of each section. For instance, we have added more information about the soluble form of CD15s (which was absent in the previous version of the manuscript), fragments about the role of CD15 in the induction of structural changes in mucins or finally about the role of myeloid-derived stem cells in the CD15-related progression of the tumor. Besides, we have described the most interesting currently undergoing clinical trials involving the assessment of CD15, which is the most important and clinically applicable information about the use of the Lewis X in the anticancer therapy. Now we hope our content is novel, interesting and thought-provoking to be acceptable. 

  1. While the title highlights the role of CD15 in anti-cancer therapy, the majority of the review describes its prognostic relevance of it in various cancers.

Response: Thank you very much for this valuable comment. We have changed the title to match the text of the review: “Prognostic and Therapeutic Role of CD15 and CD15s in Cancer”. We hope now it suits the text.

Reviewer 3 Report

    In this manuscript,  Szlasa et al. reviewed  roles of CD15 and CD15s in many types of cancers, especially with respect to cancer metastasis. They presented numerous cases in different cancer types where expression of CD15/CD15s is being used as a prognostic marker, and focused on renal cell carcinomas (RCC) to illustrate the complex relationship between expression level of CD15/CD15s and prognosis of cancer patients. They also reviewed current efforts in targeting CD15/CD15s as a form of anticancer therapy and listed clinical trials centered around CD15/CD15s.

    The authors synthesized a broad body of researches on CD15/CD15s relevance in cancers; however, there are some issues needed to be addressed before consideration for publication:

  1. The majority of the review is about expression level of CD15/CD15s and its prognosis value in different types of cancers. The authors did talk about researches on developing anticancer therapy targeting CD15/CD15s (e.g. antibodies against CD15/CD15s), but did not center their review on this topic. The review title therefore is a bit misleading and needs to change to reflect the actual focus of the review.
  2. In the abstract, the authors singled out the CD15/CD15s prognosis value in RCC and stated that high CD15 level is a good prognostic marker for RCC whereas high CD15s level relates to poor prognosis. This can be misleading, since the authors later elaborated in section 2 and 3 that in most cancer types high CD15 usually correlates to poor prognosis; and even in the case of RCC, there are many controversies around this topics and the conclusions are much more nuanced. It would be better to point out the complexity of CD15/CD15s expression level as a prognostic marker in the abstract instead of presenting it as clear cut conclusions.
  3. In section 4, starting from line 390, the authors discussed mainly about CD15/CD15s as potential markers for different cancers. In my opinion, these do not fit in as “experimental medicine” like a mAb or inhibitor. These paragraphs might be better under section 2.
  4. Besides a table of clinical trials, I think it would be more valuable if the authors can talk more in detail about one or two significant cases, and comment on their values and drawbacks.
  5. A more comprehensive conclusion section is needed. Again, I think the authors can talk about the complexity of CD15/CD15s expression level as a prognostic marker and current stage of experimental therapies targeting CD15/CD15s in a broader sense.    

Author Response

cancers-1652626

Response letter to the Reviewer #3 Report

We thank the Reviewer for encouraging feedback and appreciate the insightful comments and suggestions.

Below, we provide a point-by-point response to each of the reviewer’s comments.

All changes in the manuscript were highlighted in yellow for clarity.

We hope that the introduced revisions significantly improve the quality of this review and qualify it for further editorial stages.

Sincerely,

Authors

Major aspects:

  1. The majority of the review is about expression level of CD15/CD15s and its prognosis value in different types of cancers. The authors did talk about researches on developing anticancer therapy targeting CD15/CD15s (e.g. antibodies against CD15/CD15s), but did not center their review on this topic. The review title therefore is a bit misleading and needs to change to reflect the actual focus of the review.

Response: Thank you for this valuable suggestion. We have changed the title to better fit the content of our work: “Prognostic and Therapeutic Role of CD15 and CD15s in Cancer”.

  1. In the abstract, the authors singled out the CD15/CD15s prognosis value in RCC and stated that high CD15 level is a good prognostic marker for RCC whereas high CD15s level relates to poor prognosis. This can be misleading, since the authors later elaborated in section 2 and 3 that in most cancer types high CD15 usually correlates to poor prognosis; and even in the case of RCC, there are many controversies around this topics and the conclusions are much more nuanced. It would be better to point out the complexity of CD15/CD15s expression level as a prognostic marker in the abstract instead of presenting it as clear cut conclusions.

Response: Thank you for the comment. We have changed the abstract so that it shows the complexity of the CD15/CD15s antigens, but also states some general information about them.

  1. In section 4, starting from line 390, the authors discussed mainly about CD15/CD15s as potential markers for different cancers. In my opinion, these do not fit in as “experimental medicine” like a mAb or inhibitor. These paragraphs might be better under section 2.

Response: Thank you very much for this valuable comment. We’ve moved the paragraphs concerning the marker role of CD15 to section 2.

  1. Besides a table of clinical trials, I think it would be more valuable if the authors can talk more in detail about one or two significant cases, and comment on their values and drawbacks.

Response: Thank you very much for this valuable comment. We have divided the content of the table into two main groups (nivolumab and pembrolizumab) and described two clinical trials concerning their use in more detail. Now we hope it suits the text more.

  1. A more comprehensive conclusion section is needed. Again, I think the authors can talk about the complexity of CD15/CD15s expression level as a prognostic marker and current stage of experimental therapies targeting CD15/CD15s in a broader sense.   

Response: Thank you very much for this very valuable comment. This section has been rebuilt and expanded to include aspects of the role of CD15 as a prognostic marker, providing a more comprehensive overview.

Round 2

Reviewer 2 Report

The authors have satisfactorily improved the flow and content of the review. There are some minor corrections below that will further improve the manuscript. 

Line 64: “Can serve as” instead of “Can serves as”

Line 64/65 as well as Line 79: “high hopes are related to its use in the immunotherapy of the tumor” can be better written as “is a potentially valuable target for immunotherapy against cancer.”

1st Subheading: Please add “:” and “homeostatic” in the subheading “CD15 and CD15s: expression, structure and homeostatic function”

Line 184; Line 452; 460, 461, 462: “+” superscript for uniformity – i.e. CD15+

Line 207: replace Cd15s with “CD15s”

Please stay uniform when refereeing to the figure in the text – “Fig.1” on line 209 vs “figure 2” on line 309

Line 445 – please replace sialyl-Lex with Sialyl-Lewis X for clarity

3rd Subheading: Predictive value of CD15 and CD15s expression in various neoplasias would be better as “Prognostic significance of CD15 and CD15s expression in various neoplasms”

Line 644: There was proven should be “This was proven/shown..”

4th subheading: This should be “Experimental drugs targeting CD15”

Line 1049: “analyzis” should be “analysis”

Author Response

cancers-1652626

Response letter to the Reviewer #2 Report

We thank the Reviewer for encouraging feedback and appreciate the insightful comments and suggestions.

All changes in the manuscript were marked in different color for clarity.

We have corrected all the mentioned typos.

We hope that the introduced revisions significantly improve the quality of this review and now the manuscript may be accepted for publication.

Sincerely,

Authors

Reviewer 3 Report

The authors have answered all my questions and concerns. 

Author Response

We thank the Reviewer for the feedback.

We hope that the introduced revisions significantly improve the quality of this review and now the manuscript may be accepted for publication.

Sincerely,

Authors